# Evaluation of the Microbiome Identification of Forensically Relevant Biological Fluids: A Pilot Study

**DOI:** 10.3390/diagnostics14020187

**Published:** 2024-01-15

**Authors:** Audrey Gouello, Laura Henry, Djamel Chadli, Florian Salipante, Joséphine Gibert, Adeline Boutet-Dubois, Jean-Philippe Lavigne

**Affiliations:** 1Institut de Recherche Criminelle de la Gendarmerie Nationale, 95000 Cergy-Pontoise, France; gouello.audrey@outlook.fr (A.G.); laurahnr75@gmail.com (L.H.); chadli27@free.fr (D.C.); josephine.gibert@gendarmerie.interieur.gouv.fr (J.G.); 2VBIC, INSERM U1047, Université Montpellier, Service de Microbiologie et Hygiène Hospitalière, CHU Nîmes, 30908 Nîmes, France; adeline.dubois@chu-nimes.fr; 3Sciences Sorbonne Universtity, 75005 Paris, France; 4Aix-Marseille University, 13005 Marseille, France; 5Service de Biostatistiques, Epidémiologie, Santé Publique et Innovation en Méthodologie, Université Montpellier, CHU Nîmes, 30029 Nîmes, France; florian.salipante@chu-nimes.fr

**Keywords:** bacterial communities, biological fluids, forensics, identification, metagenomic, microbiomes, NGS sequencing

## Abstract

In forensic sciences, body fluids, or biological traces, are a major source of information, and their identification can play a decisive role in criminal investigations. Currently, the nature of biological fluids is assessed using immunological, physico-chemical, mRNA and epigenetic methods, but these have limits in terms of sensitivity and specificity. The emergence of next-generation sequencing technologies offers new opportunities to identify the nature of body fluids by determining bacterial communities. The aim of this pilot study was to assess whether analysis of the bacterial communities in isolated and mixed biological fluids could reflect the situation observed in real forensics labs. Several samples commonly encountered in forensic sciences were tested from healthy volunteers: saliva, vaginal fluid, blood, semen and skin swabs. These samples were analyzed alone or in combination in a ratio of 1:1. Sequencing was performed on the Ion Gene Studio^TM^ S5 automated sequencer. Fluids tested alone revealed a typical bacterial signature with specific bacterial orders, enabling formal identification of the fluid of interest, despite inter-individual variations. However, in biological fluid mixtures, the predominance of some bacterial microbiomes inhibited interpretation. Oral and vaginal microbiomes were clearly preponderant, and the relative abundance of their bacterial communities and/or the presence of common species between samples made it impossible to detect bacterial orders or genera from other fluids, although they were distinguishable from one another. However, using the beta diversity, salivary fluids were identified and could be distinguished from fluids in combination. While this method of fluid identification is promising, further analyses are required to consolidate the protocol and ensure reliability.

## 1. Introduction

Biological fluids comprise all liquids produced, secreted and/or excreted by a living organism. They are essential to the functioning of the human body, performing numerous vital functions such as regulating body temperature, transporting nutrients, eliminating waste, protecting against infection and lubricating organs and tissues [1] In forensics, Short Tandem Repeat (STR) genetic analyses can establish an individual’s genetic profile. However, this technique cannot determine either how the traces were deposited or the type of traces. The characterization of biological fluids is a key step in criminal investigations and can help to understand what happened at the crime scene. For example, in the case of a bite, it would be interesting to identify the suspect’s saliva, or in the case of a suspected sexual assault, it would be beneficial to highlight vaginal or semen fluids. The positioning of these biological samples on an individual or object could also imply consent/non-consent to sexual relations and support the testimony of victims and/or suspects.

All biological fluids are colonized by microorganisms living on the surface or inside the human body, forming the human microbiome. Efforts have been made to characterize this microbiome [2]. Biological fluid is currently determined using immunological, biochemical, mRNA and epigenetic methods, but these tests have sensitivity and specificity limitations [3,4,5]. Human microbiome analysis could overcome these problems. Studies have supported the use of microbiome investigation as a promising tool in forensic sciences [3]. This microbiome analysis represents a complementary tool to STR, with applications ranging from body fluid identification [6,7,8] to post-mortem interval estimation [9,10,11], sexual attack detection [12] and geographic localization [13,14,15]. However, the main concerns for forensic application are the stability and reproducibility of microbiome analysis. Its use in forensics remains difficult, especially because of the low quantity of bacterial communities in samples, the possibility to detect mixed microbiomes and the temporal variation (microbiota dynamic influenced by both external factors and host-associated factors) in the individual microbiome between body locations [16]. It is also essential to identify all bacterial communities commonly present on liquids and surfaces that could contaminate a sample [16]. Indeed, sampling requires a sufficient quantity of bacterial DNA to determine the microbiomes or at least the main bacterial phyla or genera, which are highly specific to body fluids [6,17,18]. Finally, forensic labs face numerous requests for analysis following digital penetration rape [19], rape cases or violent sexual assaults. In these cases, microbiome analyses could be a solution to detect traces of skin, semen or blood of an attacker or present in a cavity of the victim. 

While next-generation sequencing (NGS) technologies are increasingly used in most forensic laboratories to determine genetic profiles, their use for microbiome analysis needs to be developed and their place in forensics must be more clearly defined [20]. Thus, the aim of this pilot study was to assess whether analysis of the bacterial communities in isolated and mixed biological fluids could mirror the situation observed in real forensic labs. In this way, this work evaluated the possibility of identifying mixed biological fluids reproduced by the in vitro association of salivary, blood, digital, vaginal and semen samples using a metagenomic approach.

## 2. Materials and Methods

### 2.1. Volunteers and Setting 

This observational study was approved by the local ethical committee (IRCGN2021-12.03.21) and belonged to NCT04820985 for vaginal analysis. Informed consent was obtained from all volunteers before participation. A total of 30 healthy participants (22 women and 8 men) from the Gendarmerie Institute of Criminal Research (IRCGN, Cergy, France) and Nîmes University Hospital (Nîmes, France) were included in this study between January and June 2021. Non-inclusion criteria were age < 18 years old, >45 years old (for vaginal samples) and sexual abstinence ≥ 7 days (for semen samples). The following information was recorded: age, sex, reproductive status and smoking habits (tobacco or e-cigarette).

### 2.2. Sample Collection

Saliva samples were obtained from 12 participants at the IRCGN by rubbing a sterile swab around the right and left cheeks and under the tongue for 20 s in each area. Participants did not consume food, drink or smoke one hour before sampling. The volunteers had smoked approximately one cigarette pack per day for 10 and 20 years. Vaginal samples were collected from 16 consecutive gynecological adult patients (before round 1 of IVF) at Nîmes University Hospital using a sterile swab. The swabs were present in the routine biobank of the lab. The semen samples were obtained from two individuals at IRCGN after 7 days of sexual abstinence. Blood samples were collected from one male volunteer at IRCGN. Finally, skin swabs were taken from two individuals at IRCGN by sliding two sterile swabs moistened with physiological water over palms and fingers of the two hands for 1 min and then combined for the analysis. The number of semen, blood and skin samples was limited but sufficient to explore their detection in mixtures with other fluids. All samples were immediately frozen at −20 °C and stored until further investigation.

### 2.3. Sample Preparation

To assess the detection of bacterial communities in mixed fluids, all biological fluids were analyzed alone and mixed 1:1 to reflect the most commonly encountered combinations: saliva (SAL_10, SAL_11, SAL_12)/vaginal fluid (VAG_11, VAG_12, VAG_13, VAG_14), digital swab/vaginal fluid (VAG_11, VAG_15, VAG_16), digital swab/saliva SAL_10, SAL_11, SAL_12), saliva SAL_10, SAL_11, SAL_12)/semen, and vaginal fluid (VAG_11, VAG_12)/blood. This ratio was chosen to avoid any interpretation bias in the determination of bacterial communities. The different vaginal and saliva samples included in these mixtures were chosen with the aim of representing the “classical” vaginal and oral microbiomes identified in volunteers. The samples were thawed before mixing. The mixtures were performed in sterile Eppendorf^®^ tubes using sterile swabs.

### 2.4. Sample Extraction

Total microbial DNA was extracted from all samples using the Crime Prep Adem kit (Ademtech, Pessac, France) according to the manufacturer’s instructions. This kit was the same commonly used for DNA extraction in genetic profiling in France. An extraction control with ultrapure molecular biology-grade water was used. The concentrations of extracted DNA were quantified by a fluorometric approach using QUBIT^®^ instrument (Thermo Fisher Scientific, Illkirch-Graffenstaden, France), prior to metabarcoding analysis.

### 2.5. Metagenomic Investigation

The bacterial communities of extracted DNA samples were analyzed with a metabarcoding approach. Amplicon libraries were prepared according to the Metabiote^®^ solution [21], limiting bias amplifications between samples and including a positive control (artificial bacterial community) and a negative control (PCR background noise of the total library preparation). The standard range was prepared with the Ion Universal Library Quantitation Kit (Thermo Fisher Scientific) and libraries were generated using the Ion 16S Metagenomics kit (Thermo Fisher Scientific). This kit includes two sets of primers to amplify seven hypervariable regions: pool 1 targeting V2, V4 and V8 regions and pool 2 covering V3, V6–7 and V9 regions. Amplicon libraries were sequenced on a single run of Ion Gene Studio^TM^ S5 System (Thermo Fisher Scientific). After passing quality control, demultiplexing of the obtained sequences was performed by the online software Ion Reporter^TM^ 5.20.2 (Krona) provided by Thermo Fisher Scientific using the Ion 16S^TM^ Metagenomics Kit (workflow Metagenomics 16Sw1.1). The merging step or assembly of the paired-end reads was carried out using the QIIME2 (v. 2021.11) tool applying a 97% nucleic identity assembly over the entire overlap area. Sequences were aligned on the genome by sequence homology and then compared with two reference databases of ARNr 16S, Greengenes v13.5 and MicroSEQ ID v3.0. An OTU (Operational Taxonomic Unit) table was generated and relative abundance was expressed (%), corresponding to a percentage of the species representativeness per loop of the different 16S rRNA regions.

### 2.6. Statistical Analysis

Taxon abundances were calculated with PHYLOSEQ package (v. 1.40.0) and all PERMANOVA were run with vegan (v. 2.6.4) and explored by R (v. 4.2.2). The distribution of OTUs and the composition of microbial communities were analyzed by determining their relative abundance at phylum and genus levels. Each OTU was annotated on the basis of OTU clustering to obtain the species-based abundance distribution and corresponding species information. α-diversity was represented by Shannon score and relative abundance, while β-diversity was assessed using Principal coordinate analysis (PCA) with Bray–Curtis dissimilarity indices. PCA was also used to show discrimination between groups according to a selection of differentially abundant bacteria. To further determine the differences in community structure among grouped samples, statistical analysis methods such as Student’s *t*-test, Simper, Metastat, and analysis for similarities (Anosim) were used to test the significance of differences in species composition and community structure among samples. To evaluate statistical difference between individuals, statistical analyses were performed in R (v. 4.2.2). A *p*-value < 0.05 was considered significant.

## 3. Results

### 3.1. Characteristic of Volunteers

The main characteristics of participants are presented in Table 1. Twenty-two women and eight men were included, with age varying between 19 and 50 years. Four were smokers, with one being an e-cigarette smoker. One woman was pregnant.

### 3.2. Oral Microbiome Analysis

A total of 6,765,400 total mapped reads were obtained from all oral samples ranging from a minimum mapped sample read of 172,504 and a maximum of 732,955 after sequencing. Sequencing produced an average mapped sample read of 563,783. The alpha-diversity of the samples is presented in Table 2 with an average of 46,752 OTU and a Shannon index of 2.99. 

The oral microbiome of the participants was assigned into 21 phyla (Figure 1). Six major bacterial orders were observed among the various profiles: *Bacteroidales*, *Clostridiales*, *Lactobacillales*, *Neisseriales*, *Pasteurellales* and *Actinomycetales*. The data were validated by the two positive controls. Overall, age, sex and smoking habits had no influence on the results, but the relative abundance of these large bacterial orders varied between individuals, confirming an inter-individual variability. Interestingly, two profiles (SAL_2 and SAL_5) differed from the others by the identification of different bacterial orders not detected in other participants (*Enterobacteriales* for volunteer 2 and *Pseudomonadales* for volunteer 5). The beta diversity showed that the different samples were grouped except SAL_2 (Figure 2).

### 3.3. Vaginal Microbiome

A total of 3,018,174 mapped reads were obtained from all samples ranging from a minimum mapped sample read of 183,994 and a maximum of 324,706 after sequencing. Sequencing produced an average mapped sample read of 251,514. The alpha-diversity of the samples showed an average of 8182 OTU and a Shannon index of 1.02 (Table 2). Similarly to the oral microbiome, the relative abundance of these bacterial orders varied between individuals, suggesting an inter-individual variability. The majority of the vaginal samples showed a relatively high abundance of bacterial species such as *Lactobacillales*, regardless of the age of the patient (Figure 3). However, two vaginal samples (VAG_1 and VAG_10) harbored much less *Lactobacillales,* instead harboring several bacterial orders such as *Bacteroidales*, *Coriobacteriales*, *Clostridiales* and *Bifidobacteriales*. The beta diversity showed that the different samples were dispersed in the PCA (Figure 2).

### 3.4. Other Biological Fluids

We evaluated three other biological fluids: semen (*n* = 2), blood (*n* = 1), and hand cutaneous microbiomes (*n* = 2) (Figure 4). The cutaneous microbiomes harbored a high richness and diversity as expected (average mapped sample reads of 238,611 and more than 15 genera). The alpha-diversities of the different fluids varied between 3800 (blood) and 67,209 (digital skin) OTU and a Shannon index of 1.64 (blood) and 2.98 (digital skin) (Table 2). The samples showed distinct bacterial microbiome profiles between individuals, with some similarities in the detected bacterial orders (*Bacillales*, *Lactobacillales*, *Actinomycetales*, *Clostridiales* and *Rhodobacterales*). The average number of mapped sample reads of semen was 64,348. Different bacterial orders were detected in the two samples including *Lactobacillales*, *Actinomycetale*s, *Clostridiales* and *Selenomonadales*. The relative abundance of bacterial orders varied between the individuals. Finally, the average number of mapped sample reads of the blood sample was 156 with a low copy number of reads. Nevertheless, we detected a blood microbiome with diverse orders: *Enterobacteriales*, *Lactobacillales*, *Burkholderiales*, *Rhizobiales* and *Oceanospirillales.*

### 3.5. Biological Fluid Mixture Assessment

#### 3.5.1. Saliva and Vaginal Fluid Mixtures

Saliva and vaginal fluid were measured in four combinations (with four vaginal samples, VAG_11, VAG_12, VAG_13, and VAG_14, and three salivary samples, SAL_10, SAL_11, and SAL_12: two men and one woman; no smokers). The two fluids were identifiable by microbiome analysis. The typical bacterial profile of oral microbiome was detected with the presence of *Bacteroidales*, *Clostridiales*, *Neisseriales*, *Pasteurellales* and/or *Actinomycetales* (Figure 5). Moreover, the preponderance of the relative abundance of *Lactobacillales,* a marker of the vaginal microbiome present in low relative abundance in saliva, confirmed the detection of vaginal fluid. 

#### 3.5.2. Vaginal Fluid and Digital Sample Mixtures 

Vaginal fluid and cutaneous digital swab were assessed in three combinations (with three vaginal samples (VAG_11, VAG_15, VAG_16) and both digital samples). In these mixtures, a very high relative abundance of *Lactobacillales* from vaginal fluid was detected (Figure 5). The digital microbiome was particularly difficult to detect, although some microbial traits were observed with the presence of *Actinomycetales*, *Rhodobacterales*, *Pseudomonadales* or *Caulobacterales* for two combinations.

#### 3.5.3. Saliva and Digital Sample Mixtures

Saliva and cutaneous digital swabs were measured in three combinations (with three saliva samples (SAL_10, SAL_11, SAL_12) and both digital samples). Saliva/digital swab mixtures did not show the presence of specific bacterial communities from the cutaneous microbiome (Figure 5). The relative abundance of the main bacterial orders detected corresponded exclusively to the oral microbiome.

#### 3.5.4. Saliva and Semen Mixtures

Saliva and semen were mixed in three combinations (with three saliva samples (SAL_10, SAL_11, SAL_12) and both semen samples). All bacterial orders specific to both saliva and semen were identified (Figure 5). However, some bacterial orders (e.g., *Bacillales*, *Actinomycetale*s, *Clostridiales* and *Selenomonadales*) were shared by the two fluids, complicating their formal identification. It seems likely that the predominant bacterial orders detected belonged to saliva, due to the low biomass of sperm compared with saliva.

#### 3.5.5. Vaginal Fluid and Semen Mixtures

The study of bacterial communities in vaginal fluid and sperm mixtures was therefore initiated to assess the feasibility of semen identification. Vaginal fluid and semen were measured in two combinations (with two vaginal samples (VAG_11, VAG_12) and both semen samples). A large predominance of *Lactobacillales* was observed, masking the bacterial orders relative to semen (Figure 5).

#### 3.5.6. Vaginal Fluid and Blood Mixtures

Vaginal fluid and blood were measured in two combinations (with two vaginal samples (VAG_11, VAG_12) and 1 blood sample). We observed a high predominance of *Lactobacillales* in the vaginal fluid and the absence of blood-specific bacterial orders, as seen with semen (Figure 5).

#### 3.5.7. Beta Diversity among the Fluid Mixtures

The analysis of beta diversity of the fluids alone or in combination is presented in Figure 6. The oral microbiomes identified from salivary fluids alone were gathered. A clear difference can be observed with the other salivary fluids combined with other fluids. The vaginal microbiomes alone obtained from the different patients were more dispersed, notably VAG_1, VAG_2 and VAG_10. The analysis of fluid mixtures with vaginal samples showed two samples (VAG_14_SAL_10 and VAG_16_DIG_1) among the group of vaginal microbiomes alone.

## 4. Discussion

In forensic sciences, identifying the nature of a biological fluid can be important to evaluate alternative hypotheses and to reconstruct a crime scene. Our pilot study confirmed that bacterial community analysis could be assessed in routine forensic lab workflow and performed on a large panel of biological fluids (saliva, vaginal secretion, semen, blood and digital skin). We also highlighted the benefits and difficulties in identifying several mixed body fluids. In these biological fluid mixtures, the predominance of some bacterial microbiomes inhibited interpretation. Oral and vaginal microbiomes were clearly preponderant, and their presence made it impossible or difficult to detect bacterial orders or genera from other fluids, although they were distinguishable from one another.

The oral microbiomes of our 12 volunteers showed a similar core microbiome, as previously published [13,22,23]. Interestingly, two profiles (SAL_2 and SAL_5) had additional bacterial orders not detected in other volunteers (*Enterobacteriales* for volunteer 2 and *Pseudomonadales* for volunteer 5). The presence of *Enterobacteriales* in the oral microbiome is rare but sporadically observed [24]. These bacteria can be present in the subgingival plaque of subjects suffering from periodontal diseases and/or treated with antibiotics. Their appearance is facilitated by reduced salivation and saliva pH, which is responsible for an enzymatic change in the mucosa (loss of fibronectin on the surface of epithelial cells), leading to increased bacterial adherence [25]. Volunteer 5 was an e-cigarette smoker, potentially altering his oral microbiome as previously observed [13]. Chopyk et al. also showed that e-cigarettes affected oral bacterial composition with a significant increase in some species such as *Veillonella* and *Haemophilus* [26]. Moreover, cigarette use facilitates an anaerobic oral environment, leading to the depletion of aerobic bacteria [13,27]. Here, the presence of *Pseudomonadales*, an order including strict aerobic bacteria such as *Pseudomonas aeruginosa*, was intriguing. However, these groups of bacteria can colonize the oropharyngeal and respiratory epithelia, particularly dangerous in individuals with weakened immune defenses. Cigarette use induces acquired anomalies in innate and adaptive immunity, in addition to local inflammatory phenomena, thus explaining this overabundance of *Pseudomonadales* [28,29]. Park et al. also confirmed that e-cigarette use had an impact on the bacterial composition of saliva and gingival plaque. They highlighted an increase in microbial diversity and a decrease in some bacterial orders such as *Neisseriales*, *Pasteurellales*, *Actinomycetales* and *Burkholderiales*, which we also observed in the volunteer 5 [30]. In contrast, although some of the volunteers tested (SAL_3, SAL_7, SAL_9) consumed tobacco, their salivary microbiomes were relatively similar to that of non-smoking individuals. This result contradicts previous publications [13,31]. Several factors could explain these differences, such as the time between the analysis and the last cigarette smoked, the number of cigarettes smoked per day (about one cigarette pack per day for the volunteers in our study), or the volunteer’s years of smoking (between 10 and 20 years in our study). Further experiments must be performed to increase the robustness of tests on a wider panel (including more smokers with different consumption of cigarettes or e-cigarettes and with different durations) already carried out and ensure the reliability of results. However, possible inter-individual variations should be also taken into account when interpreting oral microbiome [32].

The vaginal microbiome observations were consistent with the expected results since the human vaginal microbiome is mainly composed of *Lactobacillales*, creating an acidic environment protecting against opportunistic infections or sexually transmitted diseases [6]. Interestingly, two participants (VAG_1 and VAG_10) presented vaginal samples with the presence of bacterial orders associated with the community state type (CST) IV [33] or a vaginosis, which resulted in the depletion of *Lactobacillales* [34,35] and the presence of *Gardnerella*, belonging to *Bifidobacteriales*. As the volunteers were recruited among patients in a university hospital, the results were not surprising. This raises the question in a forensic application of how to determine whether fluid is of vaginal origin when the profile reveals bacteria linked to vaginosis or another pathology. To answer this question, we need to consider both the context of the case and the nature of the sample. For example, investigations of intimate samples could easily be ascribed to these bacterial communities with vaginal fluid. For samples of unknown origin, we would need to assess whether these bacterial communities are specific to a vaginal pathology, or whether they can be identified in other human biological fluids. It appears that certain bacterial orders are not specific to vaginosis. It could be therefore assumed that detection of all these bacterial communities could confirm the bacterial profile of vaginal fluid. Vaginosis is a highly specific pathology and does not correspond to any other bacterial microbiota [36]. If all markers specific to vaginosis are detected, it may be possible to conclude that the sample is vaginal fluid. For other pathologies, this could make interpretation more complex. Finally, it is important to note that the vaginal microbiome is subject to numerous variations: menstruation, hygiene, ethnicity, contraceptive use or different sexual partners [37,38,39]. All factors that can influence the vaginal microbiota should therefore be considered in data interpretation.

Whilst the mixture of vaginal and saliva samples revealed two specific microbiomes, the other combinations tested showed less clear results. In the mixture of vaginal and digital samples, although the vaginal results were clearly predominant, some traits from the cutaneous microbiome could be observed. Moreover, in the case of unknown samples, swab samples taken from the hands and/or fingers of a suspect of rape by digital penetration (at a time several hours following the events) could be easily associated with the vaginal fluid, as digital bacteria would be at negligible levels. In the mixtures combining saliva or vaginal samples with other fluids, the predominance of oral or vaginal microbiomes prevented or limited the detection of the other microbiomes. These observations confirmed that digital, blood or semen sampling does not yield enough usable bacterial material, as previously suggested [6,18,40]. However, the use of beta diversity allowed distinguishing salivary fluids alone or in combination, suggesting an interesting direction for future development. It would be interesting to carry out an exhaustive analysis of all biological fluid mixtures (mixtures of two or more fluids) that may be encountered in forensic science. It might also be relevant to evaluate other fluid mixture ratios to determine whether identification is possible whatever the ratio tested. Other factors can influence the determination of the fluid nature, such as the complexity of the mixtures, the trace degradation, the scarcity of discriminating bacterial genera in a fluid, or the contamination of the sample by exogenous bacteria from the environment.

In forensic science, samples are often collected from mixtures, and more complex mixtures will contain more bacterial genera. For use in the field, typical bacterial profiles for each fluid need to be determined. This is difficult if these bacterial profiles are significantly different between individuals, leading to the question of identifying one or more bacterial genera specific to each biological fluid. In the future, experiments will have to define a degree of similarity between the bacterial profiles and groups of bacterial communities specific to a fluid of interest to formally identify its nature. It is also essential to consider several parameters on the interpretation of results, such as the environment, the nature of the medium sampled, the possible presence of residual microbiome, or the possible contamination of the sample [41,42]. Finally, it is also relevant to consider how to present microbiome results and to assign them statistical probability to give them probative value in court. 

Some limitations can be noted in this pilot study. To assess the detection of bacterial communities in mixed fluids, all biological fluids were analyzed mixed at a 1:1 ratio. This ratio was chosen to avoid any interpretation bias in the determination of bacterial communities, but it would be interesting to study ratios expected in real life. Moreover, the panel of biological fluid mixtures and the small number of participants were restricted in our study. However, the data obtained were comparable to previous studies on the same microbiomes [6,18,40,43]. The evaluation was performed on participants who did not eat, drink or smoke for an hour before and on semen samples only following 7 days of abstinence. These situations did not represent all samples from crime scenes. However, the aim of our pilot study was to evaluate if the detection of microbiomes composed of biological fluid mixtures was available. Moreover, we did not perform biological replicates, but the beta diversity analysis clearly clustered the different samples, suggesting a validation of our technical approach. Further experiments are needed to definitively know how the microbiome could help investigators in forensics.

## 5. Conclusions

Our pilot study demonstrated the potential and difficulties in identifying different biological fluids in cases of mixed samples by a metagenomic approach. Oral and vaginal microbiomes are particularly preponderant, and notably, the vaginal microbiome masks the other fluids, which are less rich and abundant. The use of beta diversity represents an interesting solution to discriminate the fluids alone or in combination. Numerous challenges remain, including establishing sensitive, specific and robust protocols for the tested fluids. Legal and ethical aspects of bacterial community analysis must also be addressed. It is necessary to control the nature of the data analyzed, as well as the measures for protecting, collecting, storing and disseminating these data, particularly in expert reports. 

## Figures and Tables

**Figure 1 diagnostics-14-00187-f001:**
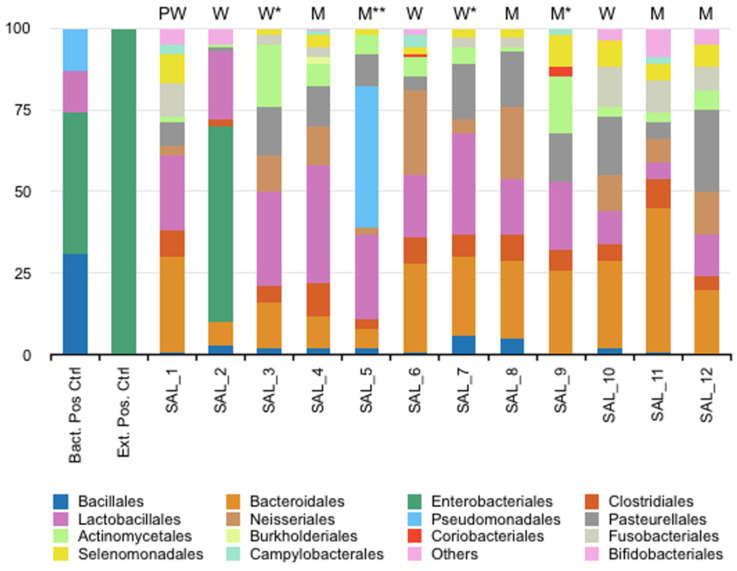
Relative abundance (%) of bacterial communities by order rank from 12 saliva samples. PW: pregnant woman, W: woman, M: man, *: tobacco smoker, **: e-cigarette smoker.

**Figure 2 diagnostics-14-00187-f002:**
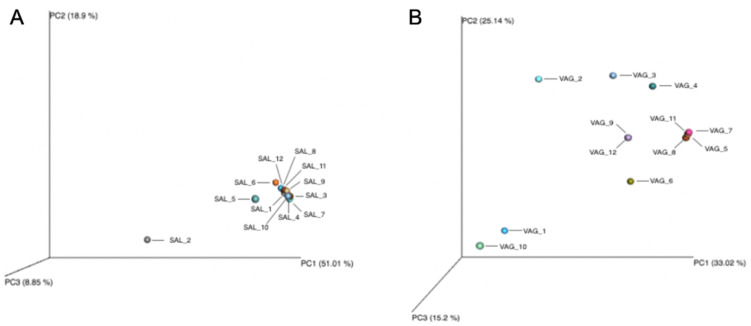
Beta diversity of the oral (**A**) and vaginal (**B**) microbiomes analyzed alone. Principal coordinate analysis (PCA) was based on the overall structure of the different microbiomes in all samples. Each data point represents an individual sample. PCA was calculated using Bray–Curtis dissimilarity.

**Figure 3 diagnostics-14-00187-f003:**
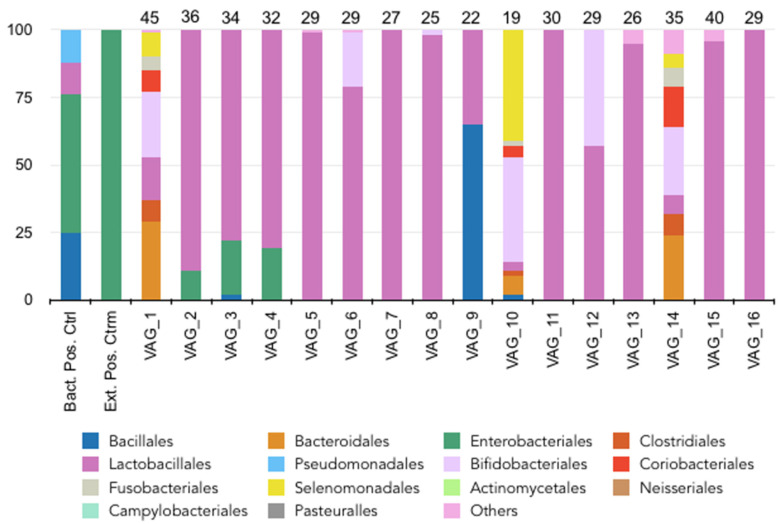
Relative abundance (%) of the main genera by order rank from 16 vaginal samples. The number above each column corresponds to the patient’s age.

**Figure 4 diagnostics-14-00187-f004:**
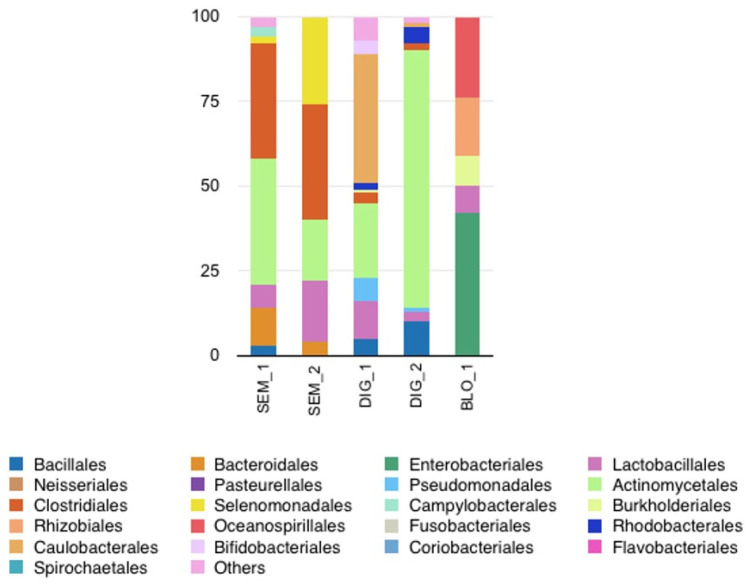
Relative abundance (%) of the main genera by order rank from different samples isolated from different body fluids: semen (SEM), blood (BLO) and digital swab (DIG).

**Figure 5 diagnostics-14-00187-f005:**
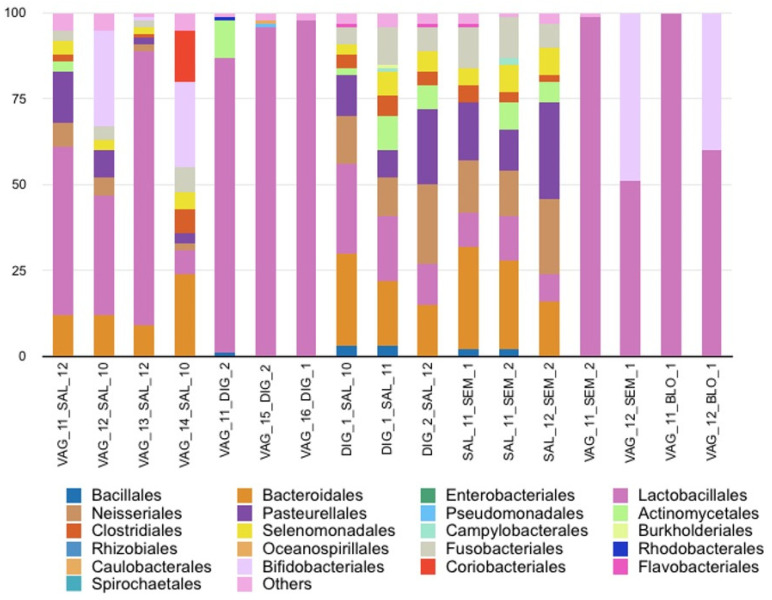
Relative abundance (%) of the main genera by order rank from different body fluid mixtures: vaginal (VAG), saliva (SAL), semen (SEM), blood (BLO) and digital swab (DIG).

**Figure 6 diagnostics-14-00187-f006:**
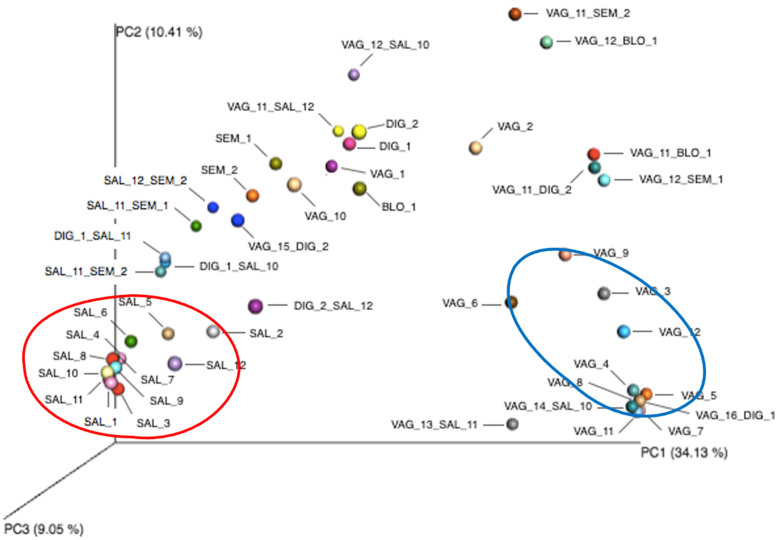
Beta diversity of the different fluids alone or in combination obtained from volunteers. Principal coordinate analysis (PCA) based on the overall structure of the different fluids’ microbiomes in all samples. Each data point represents an individual sample. PCA was calculated using Bray–Curtis dissimilarity. Gathering of oral (SAL) and vaginal (VAG) microbiomes is presented with red and blue ellipses, respectively.

**Table 1 diagnostics-14-00187-t001:** Main characteristics of participants.

Samples	Number of Volunteers				
	Saliva *n* = 12	Vaginal *n* = 16	Digital *n* = 2	Semen *n* = 2	Blood *n* = 1
Age, median (range)	35 (26–50)	29 (19–45)	37 (24–50)	37 (27–46)	50
Sex, women/men	6/6	16/0	1/1	0/2	0/1
Pregnancy	1	0	0	0	0
Smoking habits	4	0	0	0	0
E-cigarette habits	1	0	0	0	0

**Table 2 diagnostics-14-00187-t002:** Community richness and alpha diversity of the different microbiomes analyzed in the study.

	Microbiomes				
	Oral	Vaginal	Semen	Digital Skin	Blood
OTU ^1^ species, average	46,752	8182	47,311	67,209	3800
OTU species, standard deviation	14,209	1991	6732	21,536	0791
Shannon index, average	2.99	1.02	2.78	2.98	1.64
Shannon index, standard deviation	0.33	0.06	0.26	0.68	0.05

^1^ OTU, Operational Taxonomic Unit.

## Data Availability

The data that support the findings of this study are available from the corresponding author upon reasonable request.

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
