# Peer review of "Evaluation of the Microbiome Identification of Forensically Relevant Biological Fluids: A Pilot Study"

_diagnostics, 2024, doi:10.3390/diagnostics14020187_

Round 1

Reviewer 1 Report

Comments and Suggestions for Authors

The proposal and objective of the manuscript are quite interesting and important in the forensic area, where the possibility of using another type of analysis arises, in an attempt to elucidate criminal cases. However, the design of the experiments has some limitations, mainly involving the number of samples analyzed.

   When it comes to microbiome analysis, individual variables can have a considerable effect on the type of results found. Thus, it would be necessary to have a large number of individuals analyzed in order to obtain results that dilute individual factors. These limitations could be better presented and discussed in the text.

   Regarding the description of samples and experiments, adjustments are necessary:

1. Discriminate which individuals had more than one type of sample collected, since the sum of the number of participants is smaller than the sum of the samples collected.

2. In item 2.3. Sample preparation, describe which mixtures were made (despite it being included in the results) and the criteria for choosing individuals for the mixtures. Justify why not all possible combinations were carried out, as it would be interesting to discuss the possible influence of individual variability.

3. Justify why only use very few samples of semen, blood and skin. This sample number raises many doubts regarding the results. Perhaps it would be more interesting and robust to just show the results with saliva and vaginal fluid samples, performing all combinations of mixtures, since there is a larger number of samples. Or collect more semen samples, as the semen-vaginal fluid mixture is extremely important in cases of sexual crimes.

4. In Lines 187-188, the authors state that "The majority of the vaginal samples showed a relatively high abundance of bacterial species such as Lactobacillales and Bacillales". However, only 1 sample (VAG_9) shows the presence of Bacillales in Figure 3.

5. Justify why Figure 3 does not show all vaginal fluid samples. Only 12 of the 16 samples are included.

It would be important for the authors to better discuss the influence, not only of individuals, but also of the small number of blood, semen and skin samples tested and the small number of mixing experiments, on the interpretation of the results.

Author Response

Reviewer 1’s comments:

-The proposal and objective of the manuscript are quite interesting and important in the forensic area, where the possibility of using another type of analysis arises, in an attempt to elucidate criminal cases. However, the design of the experiments has some limitations, mainly involving the number of samples analyzed.

-When it comes to microbiome analysis, individual variables can have a considerable effect on the type of results found. Thus, it would be necessary to have a large number of individuals analyzed in order to obtain results that dilute individual factors. These limitations could be better presented and discussed in the text.

We agree with the reviewer. However, our study was a pilot study. Moreover:

-all the data obtained in this paper (in particular concerning OTU and alpha-diversity) were completely consistent with previous publications (see for example Ref 6, 18, 40, 43).

-the main objective of our study was to evaluate whether or not the forensic lab could detect mixed fluids in real life. Our study answers this question; even if we studied a small number of samples, our results clearly showed that certain mixtures did not allow the detection of different fluids (e.g., semen/vagina, blood/vagina).

-it was established that small-scale experiments are often useful for understanding the magnitude of variation inherent in a system. For example, a small number of samples can be selected and sequenced to shallower depth, then analyzed to determine if a larger sampling size or greater sequencing effort are required to obtain statistically significant results (see: Prosser JI. Replicate or lie. Environ Microbiol. 2010, 12 (7): 1806-1810).

-the results of beta diversity analysis in particular PCoA clearly grouped the different samples (vaginal, oral, semen, blood or skin samples are separated but each of the groups is clustered). These results confirmed the validity of our results and constituted biological replicates.

We therefore believe that our pilot study provides sufficient data to support our conclusion although we agree that extensive experiments are needed as discussed in the “limitations” section.

-Regarding the description of samples and experiments, adjustments are necessary:

  1. Discriminate which individuals had more than one type of sample collected, since the sum of the number of participants is smaller than the sum of the samples collected.

The participants corresponded to : 8 men (from IRCGN) of whom 6 donated saliva, 1 skin, semen and blood and 1 semen only; and 22 women with 16 vaginal samples (from University hospital), 6 from IRCGN of which 5 gave saliva only and 1 saliva and skin samples.

  1. In item 2.3. Sample preparation, describe which mixtures were made (despite it being included in the results) and the criteria for choosing individuals for the mixtures. Justify why not all possible combinations were carried out, as it would be interesting to discuss the possible influence of individual variability.

The different mixtures are presented. We focused on the most commonly encountered combinations in forensic procedures.We discussed this point in the limitations of our study in the Discussion section.

  1. Justify why only use very few samples of semen, blood and skin. This sample number raises many doubts regarding the results. Perhaps it would be more interesting and robust to just show the results with saliva and vaginal fluid samples, performing all combinations of mixtures, since there is a larger number of samples. Or collect more semen samples, as the semen-vaginal fluid mixture is extremely important in cases of sexual crimes.

As previously noted, this study was a pilot study (see above).

  1. In Lines 187-188, the authors state that "The majority of the vaginal samples showed a relatively high abundance of bacterial species such as Lactobacillales and Bacillales". However, only 1 sample (VAG_9) shows the presence of Bacillales in Figure 3.

We modified the sentence.

  1. Justify why Figure 3 does not show all vaginal fluid samples. Only 12 of the 16 samples are included.

These results were added in the new Figure 3.

-It would be important for the authors to better discuss the influence, not only of individuals, but also of the small number of blood, semen and skin samples tested and the small number of mixing experiments, on the interpretation of the results.

We modified the discussion accordingly.

Reviewer 2 Report

Comments and Suggestions for Authors

Summary The article evaluates the microbiome composition of biological fluids for forensic identification. The authors perform 16S rRNA species-level identification through Next-Generation Sequencing (NGS) on single-source samples and sample mixtures. They analyze 12 saliva samples, 16 vaginal samples, 2 hand swabs, 2 semen samples, and 1 blood sample from 30 healthy participants. It concludes that these samples possess distinct microbial profiles, but when mixed, the predominance of vaginal and oral bacteria hinders identification in mixtures.

Main Considerations The main limitation of this article is the extremely limited number of samples tested for three out of the five types of samples. This limits the ability to make valuable statements concerning the observed results, as they may not be representative of a larger population. Additionally, the lack of biological and/or technical replicates further limits the validity of the observed results.

Abstract

  • Lines 17-19: Mention that mRNA and epigenetic methods are also used for biological fluids identification, not just the methods described. Reference: SA Harbison & RI Fleming (2016).
  • Line 22: Revise the wording of “could be mirrored the situation…” for clarity.
  • Line 24: Clarify that skin swabs are not a fluid.
  • Lines 25-27: Note the insufficient data to support the statement, given the small number of participants for three of the fluids.
  • Line 32: Replace "distinguish" with "distinguished."

Introduction

  • Line 44: Use the singular form for “these techniques” as it refers to one method.
  • Line 49: Introduce “microbial communities” before mentioning them.
  • Lines 54-55: Suggest discussing other methods used for biological fluids identification.
  • Lines 55 and 60-61: Address the contradiction regarding the sensitivity and specificity of microbiome analysis.
  • Line 57: Use “This microbiome analysis” or “these microbiome analyses.”
  • Line 63-64: Clarify the sentence about the temporal variation of the individual microbiome.
  • Lines 66-68: Provide references for the presence of microbes in blood and semen.
  • Line 73: Clarify “be mirrored” or “mirror.”
  • Line 73-74: Use “forensic labs,” not “forensics labs.”

Materials and Methods

  • Line 89: Distinguish between liquid saliva and oral mucosa.
  • Line 91-93: Include additional information about gynaecological patients.
  • Line 93-95: Address the limited number of participants for semen, blood, and skin swabs.
  • Lines 100-101: Clarify the mixing procedure of samples.
  • Line 101: Explain the 1:1 mixing ratio for swabs and liquid samples.
  • Lines 109-110: Specify where the kit is commonly used.
  • Line 112: Use “Thermo Fisher” instead of “ThermoFisher.” Apply this change throughout the document.
  • Line 116: Provide a reference for the Metabiote solution.
  • Lines 117-118: Specify the type of artificial bacterial community used.
  • Line 123: Change “was” to “were.”
  • Lines 132-133: Clarify “species representativeness per loop.”
  • Line 135: Specify if the PHYLOSEQ package is for R.
  • Line 148: Use “P-value.”

Results

  • Line 153: Change “women” to “woman,” singular.
  • Line 156: Specify that the reference is to oral samples.
  • Line 159: Change “were” to “is.”
  • Table 2: Consider the relevance of this table.
  • Line 161: Change “analysis” to “analysed.”
  • Lines 165-166: Italicise bacterial orders.
  • Line 172: Find a better word choice for “were gathered.”
  • Figure 1: Mention if positive controls appeared as expected.
  • Figure 2: Correct the description of graphs in line 177.
  • Figure 3: Address the absence of columns for VAG_13, 14, 15, and 16.
  • Line 196: Standardize punctuation in titles.
  • Line 200: Clarify the reference to “fluids.”
  • Lines 215-217: Avoid using brackets within brackets.
  • Lines 231-232: Move the rationale explanation to the introduction.
  • Line 238: Remove extra space between “fluid-mixtures.”
  • Lines 248-251: Shift this content to the introduction.
  • Lines 256-257: This content fits better in the introduction.
  • Figure 6: Improve the quality of the red and blue oval shapes.

Discussion

  • Line 298: Clarify the reference to e-cigarette smoking.
  • Line 302: Specify the type of smoking referred to.
  • Lines 312-313: Move the smoking habits of participants to the methods section.
  • Line 314: Clarify the sentence about the wider panel.
  • Lines 320-322: Discuss other possible explanations for microbiome variations.
  • Line 325: Specify bacteria linked to vaginosis.
  • Line 330: Elaborate on bacterial orders with appropriate references.
  • Line 332-333: Provide a reference for the statement.
  • Lines 335-337: Mention this earlier to show that pathologies are not the only explanation for microbial variations.
  • Line 345: Remove double brackets.
  • Lines 348-350: Correct the reference regarding blood or semen bacterial content.
  • Lines 363-364: Consider the necessity of the sentence.
  • Lines 372-381: Address the limitations of the study and the lack of replicates. Clarify the sentence in lines 376-377. Change “with” to “of” in line 380.

Conclusions

  • Line 383: Specify the scope of demonstrating microbiome analysis in forensics.

These corrections and clarifications will enhance the accuracy and readability of the article.

Comments on the Quality of English Language

As mentioned above.

Author Response

Reviewer 2’s comments:

*Main considerations: The main limitation of this article is the extremely limited number of samples tested for three out of the five types of samples. This limits the ability to make valuable statements concerning the observed results, as they may not be representative of a larger population. Additionally, the lack of biological and/or technical replicates further limits the validity of the observed results.

We agree with the reviewer. However, our study was a pilot study. Moreover:

-all the data obtained in this paper (in particular concerning OTU and alpha-diversity) were completely consistent with previous publications (see for example Ref 6, 18, 40, 43).

-the main objective of our study was to evaluate whether or not the forensic lab could detect mixed fluids in real life. Our study answers this question; even if we studied a small number of samples, our results clearly showed that certain mixtures did not allow the detection of different fluids (e.g., semen/vagina, blood/vagina).

-it was established that small-scale experiments are often useful for understanding the magnitude of variation inherent in a system. For example, a small number of samples can be selected and sequenced to shallower depth, then analyzed to determine if a larger sampling size or greater sequencing effort are required to obtain statistically significant results (see: Prosser JI. Replicate or lie. Environ Microbiol. 2010, 12 (7): 1806-1810).

-the results of beta diversity analysis in particular PCoA clearly grouped the different samples (vaginal, oral, semen, blood or skin samples are separated but each of the groups is clustered). These results confirmed the validity of our results and constituted biological replicates.

We therefore believe that our pilot study provides sufficient data to support our conclusion although we agree that extensive experiments are needed as discussed in the “limitations” section.

*Abstract

-Lines 17-19: Mention that mRNA and epigenetic methods are also used for biological fluids identification, not just the methods described. Reference: SA Harbison & RI Fleming (2016)

We modified accordingly and the reference was added (Ref 5 in the new version of the manuscript).

-Line 22: Revise the wording of “could be mirrored the situation…” for clarity

We revised the sentence.

-Line 24: Clarify that skin swabs are not a fluid

We modified the sentence.

-Lines 25-27: Note the insufficient data to support the statement, given the small number of participants for three of the fluids

This limitation is discussed in the “Limitations” section of the manuscript. The notion of pilot study has been added throughout the text.

-Line 32: Replace "distinguish" with "distinguished."

We corrected accordingly.

*Introduction

-Line 44: Use the singular form for “these techniques” as it refers to one method.

We modified accordingly.

-Line 49: Introduce “microbial communities” before mentioning them

We modified the sentence in the new version of the manuscript.

-Lines 54-55: Suggest discussing other methods used for biological fluids identification

We modified the sentence and added a new suggested reference (Ref 5 in the new version of the manuscript).

-Lines 55 and 60-61: Address the contradiction regarding the sensitivity and specificity of microbiome analysis

We modified the sentence in the new version of the manuscript. To date, the main problem of microbiome analyses is their stability and reproducibility.

-Line 57: Use “This microbiome analysis” or “these microbiome analyses.”

We corrected the typographical error.

-Line 63-64: Clarify the sentence about the temporal variation of the individual microbiome

We modified the sentence and explained “the temporal variation”.

-Lines 66-68: Provide references for the presence of microbes in blood and semen

We added two new references (Ref 6 and 18 in the new version of the manuscript).

-Line 73: Clarify “be mirrored” or “mirror.”

We corrected accordingly.

-Line 73-74: Use “forensic labs,” not “forensics labs.”

We modified accordingly.

*Materials and Methods

-Line 89: Distinguish between liquid saliva and oral mucosa

We modified the sentence.

-Line 91-93: Include additional information about gynaecological patients

The NCT number and some information were added.

-Line 93-95: Address the limited number of participants for semen, blood, and skin swabs

We added a sentence. However we believe that this number was not a limitation in a pilot study.

-Lines 100-101: Clarify the mixing procedure of samples.

We added information.

-Line 101: Explain the 1:1 mixing ratio for swabs and liquid samples

This ratio was explained in the text. In this pilot study, our aim was to evaluate the presence of each samples present in a same ratio to avoid any interpretation biais in the microbiome analysis.

-Lines 109-110: Specify where the kit is commonly used

We added information in the new version of the manuscript.

-Line 112: Use “Thermo Fisher” instead of “ThermoFisher.” Apply this change throughout the document.

We modified accordingly throughout the manuscript.

-Line 116: Provide a reference for the Metabiote solution

The metabiote solution was firstly published in 2016. We added the reference (Ref 21 in the new version of the manuscript).

-Lines 117-118: Specify the type of artificial bacterial community used

This is a mixture provided by GenoScreen SA (Lille, France) including species or genra from Bacteroidales, Actinobacteria, Firmicutes, Proteobacteria, Acidobacteriota and Verrucomicrobiota.

-Line 123: Change “was” to “were.”

We modified accordingly.

-Lines 132-133: Clarify “species representativeness per loop.”

We clarified the sentence.

-Line 135: Specify if the PHYLOSEQ package is for R

Yes PHYLOSEQ was used and the results explored by R.

-Line 148: Use “P-value.”

We modified accordingly.

*Results

-Line 153: Change “women” to “woman,” singular.

We modified accordingly.

-Line 156: Specify that the reference is to oral samples

We modified accordingly.

-Line 159: Change “were” to “is.”

We modified accordingly.

-Table 2: Consider the relevance of this table

We proposed to maintain this Table. For the readers, it is important to quickly note the community richness of our analysis (in particular with the aim of showing that our results were comparable to other studies with more samples).

-Line 161: Change “analysis” to “analysed.”

We modified accordingly.

-Lines 165-166: Italicise bacterial orders

All bacterial orders were italicized in the submitted document.

-Line 172: Find a better word choice for “were gathered”

We corrected the sentence.

-Figure 1: Mention if positive controls appeared as expected.

Yes, the two positive controls were as expected. We added a sentence in the Results section.

-Figure 2: Correct the description of graphs in line 177

We corrected our typographical error.

-Figure 3: Address the absence of columns for VAG_13, 14, 15, and 16

These results were added in the new Figure 3.

-Line 196: Standardize punctuation in titles

This typographical error was not present in the submitted version.

-Line 200: Clarify the reference to “fluids.”

We clarified in the new version of the manuscript.

-Lines 215-217: Avoid using brackets within brackets

We modified accordingly.

-Lines 231-232: Move the rationale explanation to the introduction.

We moved the sentence in Introduction section.

-Line 238: Remove extra space between “fluid-mixtures.”

We corrected accordingly.

-Lines 248-251: Shift this content to the introduction

We moved the sentence in Introduction section.

-Lines 256-257: This content fits better in the introduction

We moved the sentence in Introduction section.

-Figure 6: Improve the quality of the red and blue oval shapes

We improved the quality of the shapes in Fig 6.

*Discussion

-Line 298: Clarify the reference to e-cigarette smoking

We modified the sentence in the new version of the manuscript.

-Line 302: Specify the type of smoking referred to

We referred to cigarettes. We modified the sentence accordingly.

-Lines 312-313: Move the smoking habits of participants to the methods section

We added information in Materials and Methods section.

-Line 314: Clarify the sentence about the wider panel

We clarified the sentence in the new version of the manuscript.

-Lines 320-322: Discuss other possible explanations for microbiome variations

We modified the sentence.

-Line 325: Specify bacteria linked to vaginosis

We modified the sentence.

-Line 330: Elaborate on bacterial orders with appropriate references

We agree with the reviewer.

-Line 332-333: Provide a reference for the statement

We added a new reference (Ref 36 in the new version of the manuscript).

-Lines 335-337: Mention this earlier to show that pathologies are not the only explanation for microbial variations.

We agree with the reviewer. However, this paragraph discusses the difficulties in interpretating microbiome data when mixed samples were detected.

-Line 345: Remove double brackets

We removed the brackets.

-Lines 348-350: Correct the reference regarding blood or semen bacterial content

We added the new references (Ref 6 and 18 in the new version of the manuscript).

-Lines 363-364: Consider the necessity of the sentence

We deleted the sentence.

-Lines 372-381: Address the limitations of the study and the lack of replicates.

The limitations are discussed as suggested by the reviewer.

-Clarify the sentence in lines 376-377. Change “with” to “of” in line 380

We modified accordingly.

*Conclusions

-Line 383: Specify the scope of demonstrating microbiome analysis in forensics

We modified the sentence.

Round 2

Reviewer 2 Report

Comments and Suggestions for Authors

Recommended corrections were carried out appropriately.